# Formulating Cybersecurity Requirements for Autonomous Ships Using the SQUARE Methodology

**DOI:** 10.3390/s23115033

**Published:** 2023-05-24

**Authors:** Jiwoon Yoo, Yonghyun Jo

**Affiliations:** DSLAB Company Ltd., Seoul 08511, Republic of Korea; yhjo@dslabcompany.com

**Keywords:** maritime cybersecurity, autonomous ships, AI security, security requirements

## Abstract

Artificial intelligence (AI) technology is crucial for developing autonomous ships in the maritime industry. Autonomous ships, based on the collected information, recognize the environment without any human intervention and operate themselves using their own judgment. However, ship-to-land connectivity increased, owing to the real-time monitoring and remote control (for unexpected circumstances) from land; this poses a potential cyberthreat to various data collected inside and outside the ships and to the applied AI technology. For the safety of autonomous ships, cybersecurity around AI technology needs to be considered, in addition to the cybersecurity of the ship systems. By identifying various vulnerabilities and via research cases of the ship systems and AI technologies, this study presents possible cyberattack scenarios on the AI technologies applied to autonomous ships. Based on these attack scenarios, cyberthreats and cybersecurity requirements are formulated for autonomous ships by employing the security quality requirements engineering (SQUARE) methodology.

## 1. Introduction

In the past few years, we have witnessed substantial growth in the development and deployment of various types of cyberphysical systems. This growth has affected nearly every aspect of our daily life, including power grids, transportation systems, medical devices, and household appliances [1]. It has also brought several changes in the shipbuilding and marine industries. With the increasing ship network connectivity, smart ships have emerged, and the race to secure autonomous ship technologies is intensifying, which may lead to a new paradigm in the maritime industry. Artificial intelligence (AI) technology, the core technology in autonomous navigation systems, is capable of processing high-dimensional information such as human cognition, learning, and reasoning based on data and knowledge [2]. AI technologies such as machine learning and deep learning are expected to support autonomous ships by processing their large amounts of generated data.

Autonomous ships are the next-generation ships that integrate advanced AI technologies, big data, and sensors; although they satisfy the anticipations of low operating costs and maritime accident prevention [3], they are exposed to many cyberthreats owing to high ship-to-land connectivity, which is, however, essential for their operation and for the management systems on land [4]. Moreover, the use of dual or backup systems should be considered to ensure that the failure of nonbackup systems does not significantly affect the ship’s ability to operate, and the ship navigation systems should be designed to ensure high operability [5]. Therefore, the communication systems that support autonomous navigation are considered an essential part of safety systems and, thus, require safety certifications [6]. To solve cybersecurity issues, various studies on ship systems and vulnerabilities in AI technology are being conducted. However, many studies focus on either ship systems or AI technologies separately and fail to consider the convergence of the two areas. The International Association of Classification Societies (IACS) has published the unified requirements (URs) for cyber-resilience in ship systems. The URs apply to all computer-based systems, such as the main-engine control, steering, cooling, fire detection, communication (including public address), and navigation systems. Moreover, the URs apply to all newly built ships contracted after 1 January 2024 but do not address the security requirements of autonomous ships.

It is known that if defects are found in the “field,” a lot of money and effort will be required for their elimination [7]; therefore, continuous efforts are being made to minimize security defects by applying security requirements at the design stage via security designs and realizing security internalization. For security internalization, security requirements must be formulated for each design stage. Security requirement engineering can be applied at the design stage to formulate security requirements from the consensus of relevant stakeholders and security experts.

This paper investigates which AI technologies are utilized in autonomous ships and analyzes those cyberthreats which may arise from the convergence of AI technologies with ship systems. Furthermore, we studied various attack scenarios to examine how previously known and researched cyberthreats can affect autonomous ships and present the necessary research direction for the cybersecurity of autonomous ships. Additionally, security requirements are formulated by applying security requirement engineering to the analyzed and investigated potential cyberthreats.

## 2. Background

### 2.1. Autonomous Ship

The core technologies of autonomous ships comprise situational recognition and detection, judgment, action and control, and infrastructure technologies, as shown in Figure 1. In autonomous ships, detection, judgment, and action are performed using AI, and only the action outcome is monitored by the crew members or coastal control centers [8].

The situational recognition and detection technology involves a vision system, such as radar, LiDAR, CCTV, and automatic identification system (AIS), that can accurately recognize weather conditions and different objects at sea.

Judgment technology allows autonomous ships to automatically sail, berth, and moor based on the data collected and predicted using vision systems. It can automatically set a safe, economic navigation route according to sea conditions, predict failures, and take action in advance.

The action and control technology controls the ship’s position, speed, and engine via AI with its judgment technology. It is a technology that allows remote control of the ship (in case of emergency) from the coastal control center.

The infrastructure technology corresponds to a seaport automation technology that can operate autonomous ships and includes laws, institutions, and standards.

Each technology is classified by its role, but in autonomous ships, these technologies and systems cannot be considered independent systems because they are recognized, detected, judged, and controlled based on each other’s data.

### 2.2. Artificial Intelligence Technology in Autonomous Ships

#### 2.2.1. Situational Recognition and Detection

According to the International Maritime Organization’s 1974 SOLAS (Chapter 5, Rule 19 of the 1999/2000 Amendment), the installation of an AIS is mandatory for most ships, except for passenger ships weighing less than 300 tons. The AIS reports static and dynamic data that are essential for the recognition of maritime traffic. The combination of AIS data with other data requires a high capability of recognition and situational judgment, and methods capable of such include neural networks, Bayesian networks, and Gaussian processes [9]. Additionally, deep-learning-based techniques that detect anomalies in maritime regions using numerous data obtained from AIS are being studied, such as variational recurrent neural networks and artificial neural networks (ANNs) [10,11]. The disadvantage of using convolutional neural networks (CNNs) to detect objects at sea is that it requires extensive computational work; as this is not always possible in a maritime environment where the size of the object in the image varies with distance, the region proposal network (RPN) is proposed as one of the ways to solve this issue. RPNs are useful when the area of an object in pixels is very small (e.g., small boats) [12].

The main challenges in recognizing a ship’s overall situation are safety and the detection of anomalies. As shown in Figure 2, an AIS, a global navigation satellite system GNSS, camera images, audio signals, and sensor data can be combined to recognize the circumstances and detect anomalies. For example, if the detected and classified results match the meta data of the AIS message, it can be considered that the situational recognition is normal. This process serves as the basis for the next step, which is to recognize the identified surrounding objects and calculate the collision probabilities using the planned path.

#### 2.2.2. Autonomous Navigation System

The autonomous operation of ships involves various AI technologies. Additionally, the structure of the automatic navigation system can be divided into four technical areas: situational awareness and detection technology via sensor data collection and analysis for autonomous operation without the intervention of human decisions; technology for making judgments such as automatic navigation, collision avoidance, and efficient route planning; technology for controlling the ship’s activities and taking action on the situations based on the judgment; infrastructure technology for operating autonomous ships from outside the ship and remote monitoring and controlling. Figure 3 depicts the structure of an autonomous navigation system [13].

Most ship collisions are caused by human accidents; they pose a major threat at sea [14]. Ship collision avoidance is regulated by the 1972 Convention on the Prevention of Collisions at Sea (COLREGs). Autonomous ships can avoid collisions with advances in deep learning. Figure 4 is a schematic representation to illustrate the collision avoidance procedure [9]. Collision avoidance involves analyzing precollected data and real-time sensor data to find actions that can follow COLREGs. The AI uses this iterative process to detect objects and implement collision avoidance procedures.

Additional decision-making support must also be possible for ship-to-ship situational awareness and collision avoidance. Research and development in deep and machine learning frameworks are needed to achieve the ship intelligence required for collision avoidance [15].

Route planning is one of the main parameters in autonomous ship systems. The purpose of route planning is to arrive at the destination with the optimal distance and time. Currently, methods such as neural networks and fuzzy and genetic algorithms are being studied for route planning in the fields of unmanned cars, mobile robots, and drones [9]. Traditional path-planning algorithms recycle historical data, resulting in poor algorithm accuracy and inefficient actual paths. As a result, research is being conducted on the routing models for autonomous unmanned ships, and improved route planning algorithms and models are under development [16,17].

#### 2.2.3. Automatic Berthing System

Berthing refers to the mooring of ships in ports. Existing large ships are supported by tugboats to help them berth. A tugboat is a marine vessel that pushes or pulls other ships with a strong driving force relative to its size. For this reason, research was previously conducted in synchronizing the movements of several tugboats. Recently, however, research has shifted to finding ways to automatically dock a ship without using additional support vessels such as tugboats.

One approach to solving the berthing problem is to use ANNs. ANN is a statistical learning algorithm inspired by biological neural networks and used in machine learning and cognitive science. The automated berthing system, using ANN, manually controls the ship to generate training data for neural network training. The ship’s status, such as the ship coordinates, direction, and distance to berthing, which is recorded during the ship steering process, is used as training data. The neural network trained from this data is used as the main controller of the berthing system. The existing ANN controllers could only dock a ship at ports where training data was collected, but recent research has suggested a controller that can automatically dock a ship in new ports without relearning the ANN controller [18,19,20]. Berthing comprises three stages: the ship’s course is changed to the optimal direction of approach for berthing, the ship’s speed decreases, and finally, the main engine stops.

The more recent learning-based methods employ deep learning. Techniques such as the deep neural network (DNN) technique, better than the conventional techniques, have been applied to improve the results of learning by increasing the hidden layers in the model for the ANN technique [21,22]. There are even studies that go beyond supervised learning and reinforcement learning and apply deep reinforcement learning, showing even superior performance with greater accuracy and efficiency [23].

## 3. Cyberattack Scenarios on Autonomous Ships

From August 2022 to March 2023, a total of 55 maritime cyber incidents were analyzed, with 28 ransomware groups, 19 unknown attack groups, and 8 APT attack groups. The primary damage types are ransomware and service disruption, accounting for more than half of all damage types. Anticipated effects and estimates for autonomous ships do not take into account the potential threats associated with cyber and cyberphysical attacks. This is because untested combinations of existing maritime systems with new autonomous navigation technologies make the comprehensive assessment of their risks and vulnerabilities impossible. Nevertheless, the interconnections among ships (and between ships and onshore infrastructure) are increasing, and as they increase, so do the potential cyberthreats [24,25].

The sensor of autonomous ships identifies the ship’s position and obstacles from the inside and outside of the ship and transmits it to the AI model. The AI model then analyzes the real-time data to determine the ways the ship can operate with stability. As a result of the analysis and judgment, the ship navigates under the actuator’s control via the control system. In other words, it collects real-time data from sensors such as AIS, GNSS, and cameras and transmits appropriate judgments (based on the analysis results) to the navigation or control system to control the ship. In this process, if there is an abnormality found in any part of the data, a chain of erroneous operations occurs. In the data flow for autonomous navigation, using the vulnerabilities in ship systems and AI technologies, various attack scenarios are presented as follows.

### 3.1. Ship Dataset Attack

If an attacker attacks the vulnerable points found at the AI learning stage, it can lead to wrong judgments owing to erroneous learning. In such cases, an attack scenario can be anticipated in which a threat is mistaken for a safe element during the ship’s voyage, and the ship is led to allow access. Jiang et al. [26] used the poisoning attack with particle swarm optimization technique to inject some malicious samples into the training dataset, resulting in a performance reduction of up to 95–33% in the worst case. In this way, poisoning attacks can be attempted on the ship identification learning as shown in Figure 5; (a) is an image of a normal ship, and (b) is an image with noise added to the existing image. To the naked eye, the images can be recognized as the same ship, but when they are injected into AI training data, the accuracy of classification can be significantly reduced [27].

Figure 6 illustrates a possible scenario when an attacker attempts a poisoning attack on the AI model training data. It is crucial for autonomous ships to undergo learning on the coast, where there are many obstacles. Through coastal navigation, the ship system learns about the environment, such as bridges, surrounding islands, and weather. In this process, a poisoning attack can occur if incorrect learning is induced in the system regarding bridges, islands, or land for malicious purposes. An AI model that learns from the erroneous data of an attacker receives real-time sensor data and makes a judgment with lower accuracy. Such misjudgments can lead to collisions as the ship deviates from its optimal or normal path and fails to identify threats. As a result, the control system may issue wrong commands, making normal operation impossible and even requiring reinstallation of the autonomous navigation system. Levine et al. [28] propose a novel approach to poisoning attack defense that, unlike traditional techniques, can be implemented using DNNs to outperform state-of-the-art attack techniques and develop an authenticated defense against common poisoning attacks.

### 3.2. Sensor Data Attack

Evasion attacks not only include a method of falsifying the objects captured by a camera but also the possibility of attacks such as spoofing and jamming by exploiting vulnerabilities in the ship system. It is possible to attack the ship system and cause malfunction of AI through infringement such as data tampering and disturbance. As shown in Figure 7, if the sensor data used as AI input data are breached by a cyberattack, safe autonomous navigation cannot be guaranteed. For example, the AIS is mandatory on many large ships, but the AIS standard does not take security into account [29].

AIS messages are not encrypted and are thus at risk of exploitation and manipulation by attackers [30,31,32]. Attacks using AIS vulnerabilities have been shown as possible in a number of scenarios, including weather information manipulation, false collision warnings, and denial-of-service attacks [30]. Data tampering and disturbance are expected to have a significant impact on unmanned autonomous navigation. As described above, spoofing, jamming, and DoS are known methods that allow the attacker–attack sensor data. If an evasion attack is used to disturb a camera or lead the sensor data in an automated berthing system to miscalculate the port distance, there is a possibility that the ship will crash into the port. Autonomous ships use many internal or external sensors, including AIS, radar, camera, SONAR, and LiDAR. Through these multiple ocean sensor data, the ship recognizes the situation and detects objects to make appropriate judgments such as course correction and collision avoidance [12,33]. Attacks that deceive or degrade sensor data can occur on the sensors used to identify ship positions and obstacles, such as the AIS, GNSS, camera, RADAR, etc. For example, GNSS spoofing sends false signals to the receiver antenna. Sensor spoofing attacks can cause errors in situational awareness and detection, leading autonomous ships to deviate from their normal course or to behave abnormally. As shown in Figure 8, errors caused by damaged sensor data lead to incorrect ship control via AI and control systems.

Autonomous driving of cars on land and autonomous aviation technology of airplanes also operates based on sensor data. Although there is some overlap in the study of autonomous navigation in cars and airplanes, the risk must be assessed considering the characteristics of autonomous ships, just as cars and airplanes have different risk profiles [34]. Software security framework, encryption, obfuscate signals, and (remote) attestation have been proposed as countermeasures in opposition to cyberthreats to sensor data, such as AIS, GNSS, and LiDAR, which are essential for ships [29,35,36].

### 3.3. Artificial Neural Network Attack

AI has evolved from classical and reactive methods to machine learning and further to deep learning, an advanced, intelligent decision-making algorithm [9]. ANN is the core technology of AI algorithms in autonomous navigation learning from data. In the case of autonomous vehicles, AI algorithms are based on neural networks, such as DNNs, CNNs, and RNNs, which are advanced forms of ANN [37]. As with other software, attacks such as brute forcing, buffer overflow, and malware injection make neural network algorithms insecure [38].

As shown in Figure 9, apart from attacking the AI model dataset, a neural network attack that attacks the AI algorithm process can undermine AI stability. Allowing neural networks to misperceive and misjudge the real-world environment can be a big problem in safety-critical applications such as autonomous navigation. An attack on ANNs is an attack on automated navigation and berthing systems, a key technology for autonomous ships. If the process of taking sensor data as input and deriving output values is compromised due to an attacker, autonomous ships will not be able to continue normal navigation.

As neural-network-based algorithms are widely used in high-stake applications, security issues have received widespread attention [39]. Securing the core technologies of autonomous navigation is equivalent to protecting not only the ship but also the environment and humans. Apruzzese et al. [40] proposed AppCon, a novel approach that aims to improve resilience against evasion attacks. Gupta et al. [41] studied the feasibility issue of well-established defense techniques against adversarial attacks and proposed guidelines for effective solutions.

### 3.4. Communication Protocol Attack

Autonomous ship systems have the potential to provide complete control to the ship in cyberattacks owing to the high network connectivity for facilitating communication with land [42,43]. As illustrated in Figure 10, a communication protocol attack is an attack in which the communication used by an autonomous ship is targeted. Various protocols are used inside and outside the ship. Examples include the Ethernet bus (which is the most commonly used land-based network), 4G/5G wireless networks, and ModBus protocol (which is often used in control systems). CAN bus, a message-based protocol widely used in vehicles, is also used. Finally, the NMEA protocol, which is used to transmit and receive sensor data on ships, is also used.

As the CAN bus and NMEA protocols used inside ships are not designed with consideration for security, there are vulnerabilities in the protocol itself. CAN bus and NMEA communication send and receive data without encryption or authentication, so if an attacker gains access to the network, all communication data can be stolen. CAN and NMEA protocols without authentication devices are vulnerable to man-in-the-middle attacks. Man-in-the-middle attacks make it possible for an attacker to manipulate or collect data and maliciously insert errors. Vulnerabilities in the satellite communications used by ships have also been reported, allowing attackers to gain access to the ship’s network [44]. For autonomous ground vehicles (AGVs) network security, Gu et al. [45] investigated the path-tracking control issue of AGVs prone to deception attacks using a learning-based Event Triggered Mechanism (ETM) and proposed a novel control strategy against deception attacks. Compared with the research covering the network security of autonomous cars, the research on the network security of autonomous ships is still underdeveloped. The automotive industry is changing its networks to Ethernet, replacing CAN. This is automotive Ethernet, and standards such as IEEE 802.3ch have emerged. The future of CAN buses in ships will change to marine Ethernet for ship navigation and communication, just as automotive Ethernet has been standardized in the automotive industry. Therefore, the contents of this study on CAN buses should continue to be researched in the ethernet environment.

## 4. Security Requirements for Autonomous Ships

### 4.1. Security Requirement Engineering

Security requirements are formulated to achieve security objectives, such as ensuring that users and applications are identified and authenticated, that only authorized data and services can be accessed, and that intrusion attempts by unauthorized users are detected. To achieve these general security goals, 12 security requirements, including identification, authentication, and authorization, can be formulated [46].

According to the analysis by Salini et al. [47], security requirements should be considered at an early stage during the SDLC process. The importance of engineering security requirements for software systems is still underestimated; they are analyzed only later in the development phase. To achieve business goals and build a software system that protects information, it is essential to identify and specify the security requirements. Therefore, a thorough analysis of security requirements is crucial for the development of software systems, which will improve the quality of development and ensure that a system is available without vulnerabilities.

The security quality requirements engineering (SQUARE) methodology employed in this study is useful for documenting and analyzing the security aspects of field systems. It also supports future improvements and system modifications. Among the security requirements engineering methodologies, it has been determined that the SQUARE methodology covers the majority of the important activities of security requirements engineering and can be used for secure software development [47]. The SQUARE methodology aims to integrate security requirements in the early stages of the development life cycle. SQUARE supports the efficient development of additional secure and survivable systems by considering security issues throughout the early stages of the system development life cycle [48]. SQUARE is performed in a total of nine steps and identifies the required inputs, suggestive techniques, and step-by-step output results. In general, the output of each step is performed sequentially as they are used as the input for the next step, but some steps can be performed in parallel [48].

We applied the SQUARE methodology to derive security requirements early on in the shipbuilding process, to apply security design, and to derive the overall requirements for autonomous navigation systems. Earlier, we identified threats to ship systems and AI while deriving four attack scenarios. The derived countermeasures are employed as artifacts to derive security requirements, and the countermeasures derived from the attack scenarios are leveraged to set security goals.

### 4.2. Derive Security Requirements

#### 4.2.1. Agreeing on Definitions

The first task is to agree on a common set of security definitions. They are to be defined by considering an organization’s security policy and vision. Given the differences in the expertise and experience between security requirement engineers and stakeholders, any term can carry different definitions among participants. Therefore, these differences in perspective must be resolved. The cyber-resilience of ships (IACS UR E26) and cyber-resilience of onboard systems and equipment (IACS UR E27), which describe the minimum requirements for cyber-resilience of ships, define some of the terms, as shown in Table 1. After finalizing the definition of each term, a complete glossary list is defined and shared with the stakeholders.

#### 4.2.2. Identify Security Goals

Autonomous ships face potential cyberthreats in AI, sensor data, networks, and more, which can allow cyberattacks to succeed. In Section 3, we presented cyberattack scenarios for autonomous ships. Through these, the following five security objectives were established.

Integrity: Ensure the integrity of sensor data.Reliability: Ensure the reliability of sensor data.Stability: Ensure the stability of AI.Access control: Access the control of the ship’s network.Availability: Ensure the availability of autonomous navigation systems.

#### 4.2.3. Develop Artifacts

In the existing design process of a ship, each design drawing is produced through the process of conceptual, basic, and detailed designs. In this study, autonomous navigation systems and components were identified via case studies of autonomous ships rather than through the original design process. As shown in Figure 11, the attack path of the attacker in the network modeling was identified using the attack scenarios for an autonomous ship devised earlier via the network logical modeling [49]. Cyber attackers may attempt to attack datasets and neural networks in Zones 1 and 2 and sensor data and protocols in Zones 3 and 4. In each attack scenario, the vulnerable attack vectors for autonomous ships include AI models, sensors, neural networks, and weak protocols.

The attack scenarios devised in Section 3 are AI model attack methods that do not occur on existing ships but are possible on autonomous ships. As shown in Figure 12, Attacks highlighted in orange will be spoofing on sensor data as jamming further increases in autonomous ships due to the increased connectivity.

Additionally, to replace misuse cases, cyberattack scenarios for autonomous ships were devised as follows Table 2 through cyberthreats on ship systems and AI.

#### 4.2.4. Perform Risk Assessments

According to onboard use and application of computer-based systems (IACS UR E22), autonomous navigation systems of autonomous ships can be classified as Category III with respect to the general classification of system capabilities or the degree of impact of malfunctions and the sensors and controllers below the degrees in Category III can be classified as Category II. In the attack tree shown in Figure 12, cyberthreats on autonomous ships can be classified as follows. Referring to the recommendation on incorporating cyber risk management into safety management systems (IACS No. 171), the threats were assessed considering their impact on autonomous ships and attacker accessibility. Reflecting the classification of IACS UR E22 according to system capabilities, the threats were classified on a scale of 1–3, as shown in Table 3.

In this study, risk analysis and evaluation were performed by applying an informal approach among many risk analysis methodologies. The process adopted the ATR model [50], an asset-based method, to conduct a risk assessment with a focus on assets and threats, as shown in Table 4.

#### 4.2.5. Select Elicitation Technique

Although SQUARE introduced several extraction techniques suitable for deriving requirements, instead of using the extraction techniques introduced by SQUARE in this study, we used a scenario-based requirements analysis method to formulate security requirements based on the analyzed potential threats and attack scenarios. Scenario-based requirements analysis is a method of describing system requirements through scenarios extracted from real-world experience or accessible examples. Among the scenario-based requirements analysis methodologies, the SCRAM methodology is based on the following four techniques [51].

Prototype or concept demonstration-Core concepts provide design artifacts that can be reproduced by the user.Scenario-Designed artifacts allow users to relate their work to the design in the user environment.
Design rationale-The designer’s reasoning is open to the user to encourage user participation in the decision-making process.Requirement priorities identification

The above techniques are combined with the requirements formulation process, and the final method comprises the following steps:Identification of initial requirements and domain analysis;Description or concepts and property setting;Derivation and analysis of requirements;Validation of requirements.

#### 4.2.6. Elicit Security Requirements

Potential threats of autonomous ships were analyzed to devise various attack scenarios. To formulate security requirements for attack scenarios and security risks, security requirements derivation techniques and other artifacts were used to formulate security requirements in terms of what the autonomous navigation systems should do.

R-01. The sensor’s function for verifying the identification data should be provided.R-02. Sensor data transmission must be protected against tampering.R-03. In the event of a denial-of-service attack, minimum functions for the operation of the ship should be provided.R-04. Security design must be performed to eliminate defects in the neural network.R-05. An encryption mechanism should be used to protect communication data.R-06. AI must be trained using reliable training data.R-07. Unauthorized network connections should be blocked.R-08. Protective measures should be implemented to prevent malware.

#### 4.2.7. Categorize Requirements

The initially formulated requirements are classified, and naming and categorization are performed. Formulated requirements are grouped, and a unique name is created for the group to list the requirements. Table 5 groups the requirements into security objectives such as availability, integrity, reliability, stability, and access control.

#### 4.2.8. Prioritize Requirements

The system category of the IACS UR E22 classifies systems in which failure of the system immediately poses a threat to humans, ship safety, and the environment as Category III, the highest class. The lowest class, Category I, is classified as a system in which system failure does not pose a threat. Therefore, the requirement priority is determined based on high availability. According to the system categorization in E22 and the results of risk assessment, the requirements are prioritized, as shown in Table 6.

#### 4.2.9. Requirements Inspection

The devised requirements were mapped onto the IACS requirements, as shown in Table 7.

Each requirement of IACS UR E27 follows IEC 62443-3-3, which describes the security requirements for industrial control systems. IEC 62443-3-3 defines a security level (SL) from SL-0, which does not require cybersecurity, to SL-4, which requires a strong cybersecurity level, as shown in Table 8.

Among the UR E27 requirements, requirements such as identification and authentication of software processes and devices and multifactor authentication are additional security requirements to meet a minimum SL-2 security level. The DNV classification divides security levels into stages, and cybersecurity requirements for autonomous ships are the highest level, the “Cyber secure ADVANCED (+)” level. This level is equivalent to the SL-4 level, the highest security level defined by IEC 62443-3-3.

Although the security requirements formulated in this study differ from those required by the IACS, they require the same basic requirements and additional requirements as those set forth by the IACS. The DNV classification also requires autonomous ships to meet strong security requirements. Therefore, the derived security requirements are the minimum-security requirements to combat the potential cyberthreats identified earlier.

## 5. Conclusions

Ship sensor data affects the AI perception, judgment, and control, and AI, in turn, affects the ship control. In autonomous ships, individual systems are not independent; each system plays its own role and generates data that affects other systems. Cybersecurity threats can also affect not just one system but the entire autonomous ship. Autonomous ships, as anticipated, reduce operating costs and maritime accidents, and if an accident is caused due to a cyberattack on an autonomous ship, it will be difficult to achieve its goals. Hardenings, such as logical access control, separation of duties, and enhanced account security, should be relatively easy to apply. However, if hardening cannot address the problem due to the nature of ships, even if a security defect is discovered, a great deal of effort and money are necessary to take the appropriate measures. Therefore, to prevent cyberattacks, each component of the ship system must be designed so that they do not affect other systems during cyberbreaches.

In this study, we investigated the vulnerabilities and threats in ship systems and AI technologies. We also investigated and analyzed potential cyberthreats to the AI technologies used on autonomous ships. As autonomous ships are still in the research stage, more advanced systems will, perhaps, be applied during the commercialization phase. The convergence of existing ship systems with AI technology also poses the possibility of new unidentified threats. Other studies pertaining to cyber risk and security requirement derivation for autonomous ships [4,29] are limited to ship systems. In order to respond to new threats, we have presented attack scenarios utilizing currently existing threats when fused alongside artificial intelligence technology and applied security requirement engineering to derive minimum security requirements for autonomous ships. Our work provides motivation to think differently of the research direction of autonomous ship cybersecurity.

In recent years, international organizations such as IMO and IACS have emphasized the application of cybersecurity on ships. Autonomous ships will require the same or higher level of cybersecurity than is currently required. Therefore, there is a necessity for cybersecurity research on to-be-applied AI technologies and autonomous navigation systems based upon a comprehensive understanding of the maritime cyber risk environment. Through cooperation and research between stakeholders in the shipbuilding and marine industries and the security industry, security requirements that must be applied in the design stage should be devised to ensure the safety of autonomous ships. It is believed that applying ship-specific penetration testing and risk assessment methods will lead to more in-depth security requirements. We look forward to the internalization of cybersecurity in autonomous ships by means of our research.

## Figures and Tables

**Figure 1 sensors-23-05033-f001:**
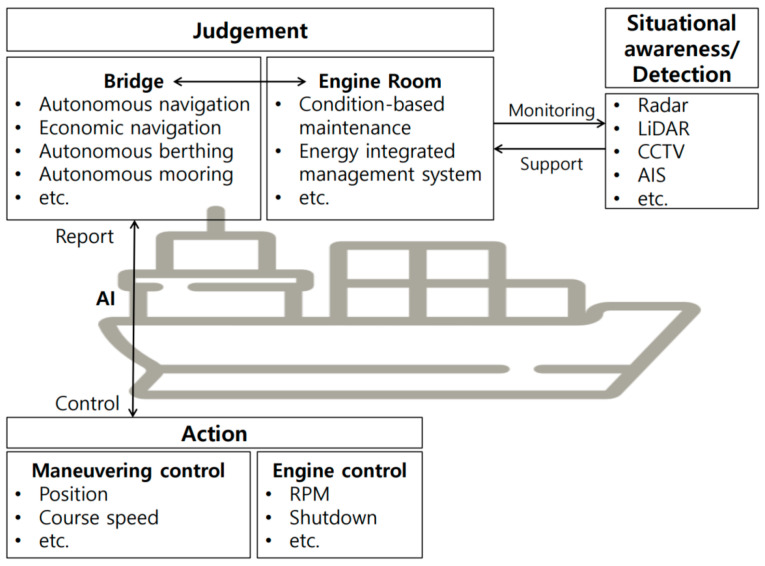
Systems in autonomous ships.

**Figure 2 sensors-23-05033-f002:**
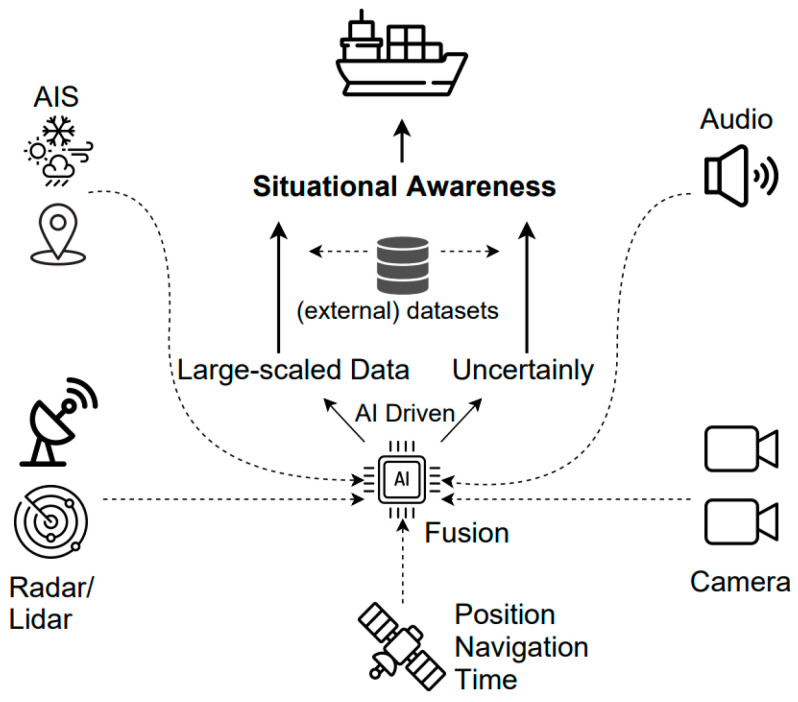
Overall view of an AI-driven maritime situational awareness system.

**Figure 3 sensors-23-05033-f003:**
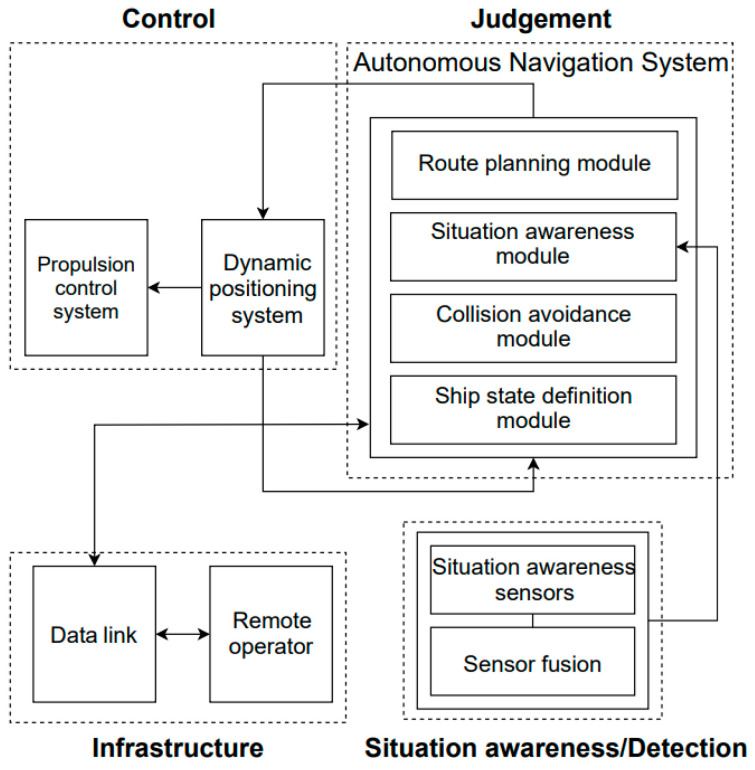
Autonomous navigation system (ANS) architecture.

**Figure 4 sensors-23-05033-f004:**
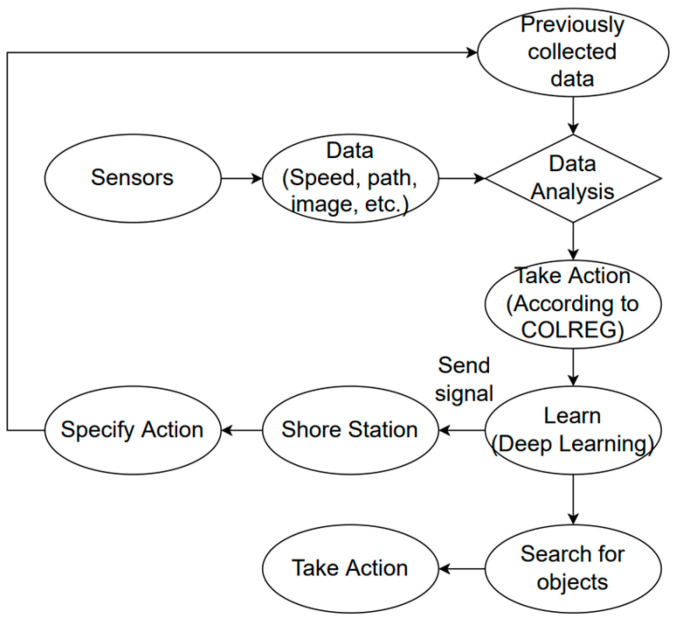
Schematic representation of autonomous collision avoidance.

**Figure 5 sensors-23-05033-f005:**
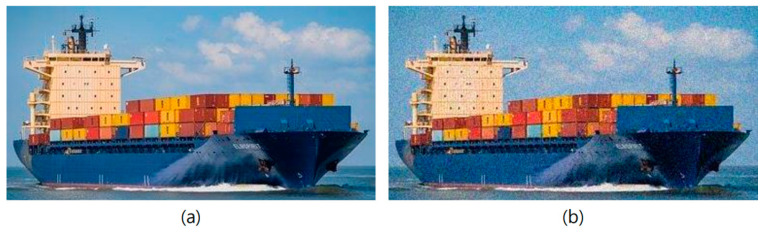
(**a**) Dataset sample (**left**) and (**b**) modified dataset sample (**right**) for a poisoning attack.

**Figure 6 sensors-23-05033-f006:**
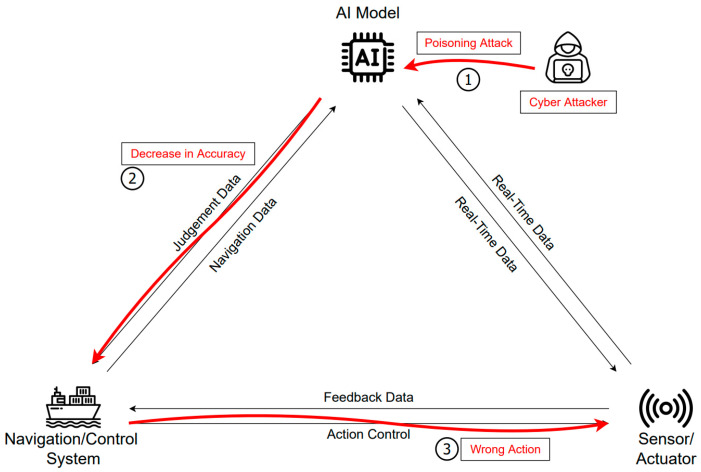
Ship dataset for a poisoning attack scenario.

**Figure 7 sensors-23-05033-f007:**
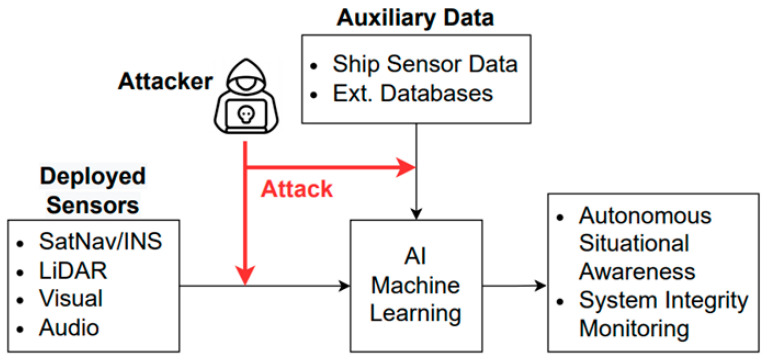
Cyberattack scenario of identification data.

**Figure 8 sensors-23-05033-f008:**
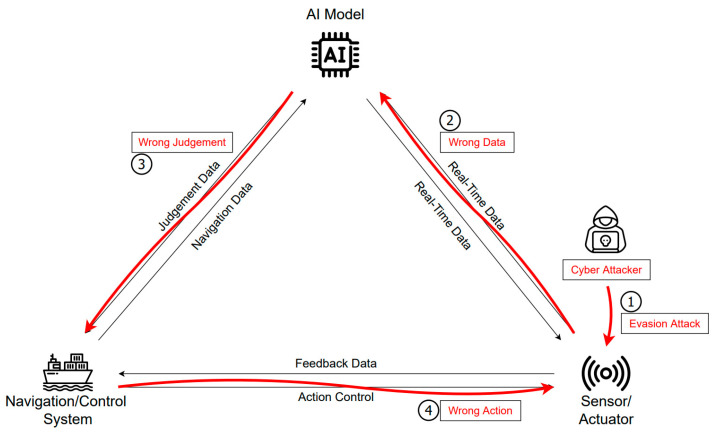
Attack scenario to deceive or degrade sensors.

**Figure 9 sensors-23-05033-f009:**
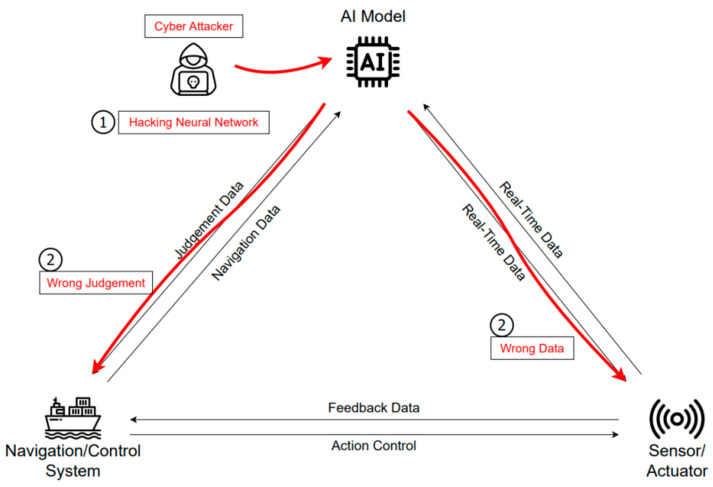
Scenario of an attack on a neural network.

**Figure 10 sensors-23-05033-f010:**
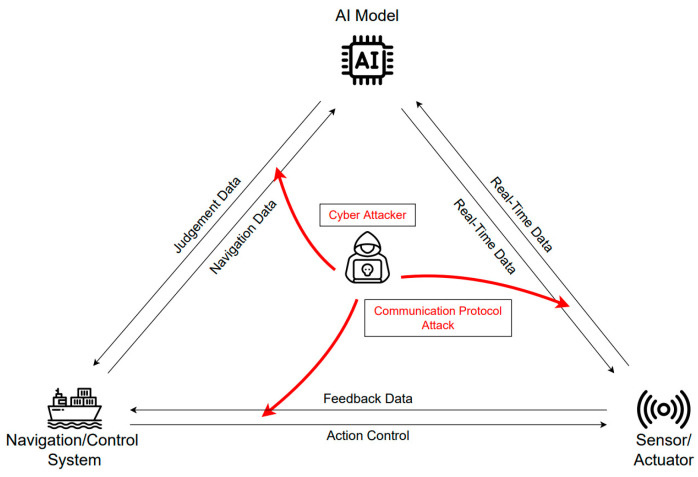
Communication protocol attack.

**Figure 11 sensors-23-05033-f011:**
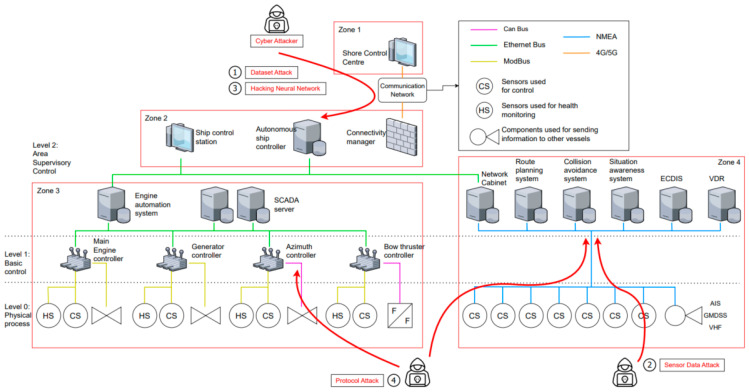
Cyber attack route.

**Figure 12 sensors-23-05033-f012:**
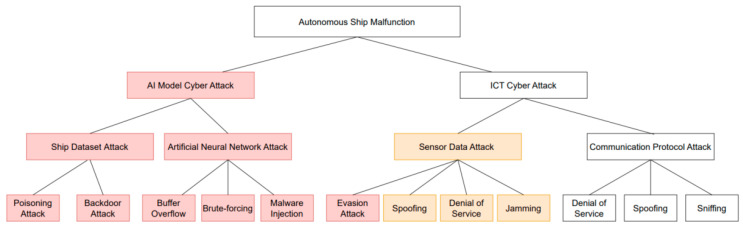
Autonomous ship attack tree.

**Table 1 sensors-23-05033-t001:** Sample of terms and definitions.

Term	Definition
Attack surface	The set of all possible points where an unauthorized user can access a system and extract data. The attack surface comprises two categories: digital and physical. The digital attack surface encompasses all the hardware and software that connect to an organization’s network. These include applications, code, ports, servers, and websites. The physical attack surface comprises all endpoint devices that an attacker can gain physical access to, such as desktop computers, hard drives, laptops, mobile phones, removable drives, and carelessly discarded hardware.
Authentication	Provision of assurance that a claimed characteristic of an entity is correct.
Compensating countermeasure	An alternate solution to a countermeasure employed in lieu of or in addition to inherent security capabilities to satisfy one or more security requirements.
Computer-based system (CBS)	A programmable electronic device, or interoperable set of programmable electronic devices, is organized to achieve one or more specified purposes, such as collection, processing, maintenance, use, sharing, dissemination, or disposition of information. CBSs onboard include IT and OT systems. CBS may be a combination of subsystems connected via a network. Onboard CBSs may be connected directly or via public means of communication (e.g., Internet) to ashore CBSs, other vessels’ CBSs, and/or other facilities.

**Table 2 sensors-23-05033-t002:** Potential cyberthreat and attack scenario in autonomous ships.

Category	Potential Threat
Systems in ship	Spoofing Denial of service Jamming Obfuscation Covert channels attacks Steganography Weak encryption Authentication Firmware update Data corruption
AI	Poisoning attack Evasion attack Inversion attack Model extraction attack Buffer overflow Brute forcing Malware injection
Attack scenario	Ship dataset attack Sensor data attack Artificial attack Communication protocol attack

**Table 3 sensors-23-05033-t003:** Threat categories.

No.	Threat List			Value
	Division 1	Division 2	Division 3	
1	AI model cyber attack	Ship dataset attack	Poisoning attack	2
2			Backdoor attack	2
3		Sensor data attack	Evasion attack	3
4			Spoofing	3
5			Denial of service	3
6			Jamming	2
7		Artificial neural network attack	Buffer overflow	3
8			Brute forcing	3
9			Malware injection	2
10	ICT cyber attack	Communication protocol attack	Denial of service	3
11			Spoofing	3
12			Sniffing	1

**Table 4 sensors-23-05033-t004:** Risk assessment results.

Level	Risk
High	Trick sensor’s identification data to cause AI malfunction Tampering sensor data to cause AI malfunction Denial-of-service attacks to block transmission of sensor data Buffer overflow or brute forcing to paralyze AI functions DoS attack on communication protocols to block internal network communication Tampering with the data transmitted over vulnerable protocols
Medium	AI malfunction or performance degradation due to contaminated training data AI malfunction due to a backdoor attack on training data AI malfunction due to sensor data disturbance AI function paralysis due to malware injection into an artificial neural network
Low	Tapping data transmitted over insecure protocols

**Table 5 sensors-23-05033-t005:** Grouped security requirements.

Group	Code	Requirements
Integrity	R-02	Sensor data transmission must be protected against tampering.
	R-05	An encryption mechanism should be used to protect communication data.
Reliability	R-01	The sensor’s function of verifying the identification data should be provided.
	R-04	Security design must be devised to eliminate defects in the neural network.
	R-05	An encryption mechanism should be used to protect communication data.
Stability	R-04	Security design must be devised to eliminate defects in the neural network.
	R-06	Artificial intelligence must be trained using reliable training data.
	R-08	Protective measures should be implemented to prevent malware.
Access control	R-07	Unauthorized network connections should be blocked.
	R-08	Protective measures should be implemented to prevent malware.
Availability	R-03	In the event of a denial-of-service attack, the minimum functionality for the operation of ships must be provided.

**Table 6 sensors-23-05033-t006:** Prioritize requirements.

Level	Code	Requirement
High	R-02	Sensor data transmission must be protected against tampering.
	R-03	In the event of a denial-of-service attack, the minimum functionality for the operation of ships must be provided.
	R-04	Security design must be performed to eliminate defects in the neural network.
	R-05	An encryption mechanism should be used to protect communication data.
	R-08	Protective measures should be implemented to prevent malware.
Medium	R-01	The sensor’s function of verifying the identification data should be provided.
	R-06	Artificial intelligence must be trained using reliable training data.
	R-07	Unauthorized network connections should be blocked.
Low	R-02	Sensor data transmission must be protected against tampering.

**Table 7 sensors-23-05033-t007:** Requirements mapping with E27.

Requirements	IACS UR E27 Control
R-01	Software process and device identification and authentication
R-02	Communication integrity
R-03	Denial-of-service protection
R-04	Security functionality verification
R-05	Communication integrity
R-06	Input validation
R-07	Network and security configuration settings
R-08	Malicious code protection

**Table 8 sensors-23-05033-t008:** IEC 62443-3-3 security level.

Security Level	Requirement
0	Does not require security specifications or protections
1	Requires protection against unintended incidents
2	Requires protection against intentional incidents perpetrated with simple means, few resources, basic knowledge, and low motivation
3	Requires protection against intentional incidents perpetrated with advanced means, sufficient resources, average knowledge, and medium motivation
4	Requires protection against intentional incidents perpetrated with very advanced means, major resources, advanced knowledge, and high motivation

## Data Availability

Not applicable.

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
