# Peer review of "Formulating Cybersecurity Requirements for Autonomous Ships Using the SQUARE Methodology"

_sensors, 2023, doi:10.3390/s23115033_

Round 1
Reviewer 1 Report
This study presented possible cyberattack scenarios on the AI technologies applied to autonomous ships. Based on these attack scenarios, cyberthreats and cybersecurity requirements were formulated for autonomous ships by employing the security quality requirements engineering methodology. However, this paper seems more like a review paper while not a research paper.
1) This paper seems more like a review paper. The authors must clearly show the difference and improvements in comparison with the existing results in the view of technique analysis.
2) The motivation on why to propose such a framework and strategy in real-world applications should be clearly emphasized. It would be much better if some guideline remark words on practical applications should be given.
3) Update the recent reference related to this work; Path tracking control of autonomous vehicles subject to deception attacks via a learning-based event-triggered mechanism, in ieee transactions on neural networks and learning systems.
4) Some examples are needed to be further expanded and including some remarks to show the effectiveness and efficiency of the proposed method, compared with others.
This study presented possible cyberattack scenarios on the AI technologies applied to autonomous ships. Based on these attack scenarios, cyberthreats and cybersecurity requirements were formulated for autonomous ships by employing the security quality requirements engineering methodology. However, this paper seems more like a review paper while not a research paper.
1) This paper seems more like a review paper. The authors must clearly show the difference and improvements in comparison with the existing results in the view of technique analysis.
2) The motivation on why to propose such a framework and strategy in real-world applications should be clearly emphasized. It would be much better if some guideline remark words on practical applications should be given.
3) Update the recent reference related to this work; Path tracking control of autonomous vehicles subject to deception attacks via a learning-based event-triggered mechanism, in ieee transactions on neural networks and learning systems.
4) Some examples are needed to be further expanded and including some remarks to show the effectiveness and efficiency of the proposed method, compared with others.
Author Response
We appreciate your invaluable feedback. We revised our manuscript based on the critique we were given and feel that the manuscript is much improved as a result. Through the attachment, you will find that we itemized the issues and outlined our responses to each reviewer’s detailed comments and reflected the changes in our revised manuscripts.

Reviewer 2 Report
1-The paper is a survey and doesn't have any contributions.
2- There is no applying for AI model to detect the attacks.
Author Response

(The authors gave the same response as above.)

Reviewer 3 Report
Review
Formulating cybersecurity requirements for autonomous ships 2
using the SQUARE methodology
The authors have produced a well written paper that clearly states its objectives and findings. I think this will be a valuable piece of work for an audience in autonomous vehicles and IoT, multi agent systems, etc.
I would like to encourage the authors to make some minor improvements before publication to enhance the value of the paper.
1. There are many diagrams, some are repeated. I would pick one diagram to represent 5 and 7 and 9 for brevity.
2. The authors mention the CANbus is used on ships. This seems increasingly old fashioned. Modern vehicles are unlikely to use something like the CANbus. What does the future look like?
3. The authors mention risk analysis - this is more risk classification. What about mitigating strategies. There isn't much mention of what can be done about these vulnerabilities. Some words on this would make the paper much stronger. Section 4.2 has some words, but these are quite obvious "fix it" remarks. Is there something more systematic / systemic that is particular to shipping here? The authors focus on sensors as a weakness. What would be the role of redundancy, for example?
4. A ship is a human-technology interface with many people (unlike a car which has mainly one person at the wheel). What are the security and automation implications of the human factors?
5. Figure 12 is pretty but it doesn't convey any useful information. It needs more words of explanation and the text in the figure is too small to read easily.
6. In table 2, how is steganography a relevant issue?
7. Who are the likely attackers of a vessel such as a ship? Who stands to gain from infiltrating these systems? Terrorists? Industrial espionage? Some words about this would be helpful in understanding the attach profile.
8. In the conclusions the authors point out (rightly) that ships are expensive systems to build and to run. It would be nice to see a few words on what kinds of changes can be implemented for existing ships while in service. AI still feels like a subject of marginal relevance to most of the floating systems in use.
I hope these remarks are helpful.
Author Response

(The authors gave the same response as above.)

Round 2
Reviewer 2 Report
NA
NA